# Cyto-Genotoxic and Transcriptomic Alterations in Human Liver Cells by Tris (2-Ethylhexyl) Phosphate (TEHP): A Putative Hepatocarcinogen

**DOI:** 10.3390/ijms23073998

**Published:** 2022-04-03

**Authors:** Quaiser Saquib, Abdullah M. Al-Salem, Maqsood A. Siddiqui, Sabiha M. Ansari, Xiaowei Zhang, Abdulaziz A. Al-Khedhairy

**Affiliations:** 1Zoology Department, College of Sciences, King Saud University, P.O. Box 2455, Riyadh 11451, Saudi Arabia; alsalem1985@hotmail.com (A.M.A.-S.); maqsoodahmads@gmail.com (M.A.S.); kedhairy@ksu.edu.sa (A.A.A.-K.); 2Botany and Microbiology Department, College of Sciences, King Saud University, P.O. Box 2455, Riyadh 11451, Saudi Arabia; sabiha.mahmood003@gmail.com; 3State Key Laboratory of Pollution Control & Resource Reuse, School of the Environment, Nanjing University, Nanjing 210023, China; zhangxw@nju.edu.cn

**Keywords:** organophosphorus flame retardants, TEHP, genotoxicity, hepatotoxicity, toxicogenomics, apoptosis, transcriptomics, oxidative stress, DNA damage, cytotoxicity

## Abstract

Tris (2-ethylhexyl) phosphate (TEHP) is an organophosphate flame retardant (OPFRs) which is extensively used as a plasticizer and has been detected in human body fluids. Contemporarily, toxicological studies on TEHP in human cells are very limited and there are few studies on its genotoxicity and cell death mechanism in human liver cells (HepG2). Herein, we find that HepG2 cells exposed to TEHP (100, 200, 400 µM) for 72 h reduced cell survival to 19.68%, 49.83%, 58.91% and 29.08%, 47.7% and 57.90%, measured by MTT and NRU assays. TEHP did not induce cytotoxicity at lower concentrations (5, 10, 25, 50 µM) after 24 h and 48 h of exposure. Flow cytometric analysis of TEHP-treated cells elevated intracellular reactive oxygen species (ROS), nitric oxide (NO), Ca^++^ influx and esterase levels, leading to mitochondrial dysfunction (Δ*Ψm*). DNA damage analysis by comet assay showed 4.67, 9.35, 13.78-fold greater OTM values in TEHP (100, 200, 400 µM)-treated cells. Cell cycle analysis exhibited 23.1%, 29.6%, and 50.8% of cells in SubG1 apoptotic phase after TEHP (100, 200 and 400 μM) treatment. Immunofluorescence data affirmed the activation of P53, caspase 3 and 9 proteins in TEHP-treated cells. In qPCR array of 84 genes, HepG2 cells treated with TEHP (100 µM, 72 h) upregulated 10 genes and downregulated 4 genes belonging to a human cancer pathway. Our novel data categorically indicate that TEHP is an oxidative stressor and carcinogenic entity, which exaggerates mitochondrial functions to induce cyto- and genotoxicity and cell death, implying its hepatotoxic features.

## 1. Introduction

The global consumption of flame retardants (FRs) in 2019 has surpassed 2.9 million tons, out of which 18% was shared by organophosphate flame retardants (OPFRs). Based on the IHS Consulting 2020 survey, it was anticipated that between 2019 and 2025, the use of FRs will increase worldwide at an annualized rate of 2.7% [1]. OPFRs are profoundly added to various consumer and industrial products such as electronics, furniture, textiles, plastics, building materials and vehicle parts [2]. OPFRs have the tendency to erode from the products, resulting in their appearance in drinking water, food, outdoor and indoor air, waste water, sludge, sediment, surface water, and indoor dust [3]. Abrasion, leaching, and volatilization are the prominent processes behind the release of OPFRs from the products [4,5]. Environmental studies have affirmed the presence of OPFRs (1600 ng/L) in a sample of drinking water in Korea [6]. Indoor air and dust samples collected from New York, USA have been detected for 2.96 to 635 ng/m^3^ and 16.2 to 224 μg/g of OPFRs [7]. Soil and river sediment samples collected from Tianjin, China and Bagmati River, Nepal showed 37.7 to 2100 ng/g and 983 to 7660 ng/g of OPFRs [8,9]. Persistent use of OPFRs in consumer products and their appearance in environmental samples raises concerns for human exposure via the consumption of drinking water, inhalation of indoor dust, and absorption of contaminated food [10]. Human biomonitoring studies have also confirmed the presence of OPFRs in human body fluids [11,12].

Tris (2-ethylhexyl) phosphate (TEHP) is an OPFRs and plasticizer, mostly used in polyvinyl chloride and other polymers. Significant levels of TEHP have been detected in marine sediments from coastal areas of Columbia and San Francisco Bay [13,14]. A human non-small cell lung cancer cell line (NCI-H1975) exposed to TEHP (0–200 μM) for 72 h demonstrated cytotoxicity, oxidative stress and apoptosis [15]. In vitro studies using two-hybrid gene reporter and e-screening assays have demonstrated the antiestrogenic properties of TEHP [16]. Indoor air dust analyzed for the presence of OPFRs exhibited higher concentration of TEHP [17]. Elderly persons exposed to TEHP via inhalation of indoor dust were frequently detected with higher concentrations in the blood [17]. Hair from Norwegian mothers and children’s exposed to indoor dust exhibited TEHP in 93% of samples, although the level was relatively higher in mothers than in children’s [18]. Distantly, elevated levels of TEHP has been reported in processed and non-processed foods [19]. Meat samples (chicken, beef, and pork) in China were detected with a TEHP mean concentration of 3.07 ng/g dw [20]. Dietary intake of TEHP containing rice and vegetables has been attributed as the main factor for exposure in male and female children in China [20]. Human breast milk has been detected with 0.26–2.1 ng/mL of TEHP; it is estimated that an infant (0–1 year) consuming 450 mL daily would result in average daily intake of 76 ng/kg b.w. of TEHP [11]. Consequently, dietary intake of TEHP has been recognized as one of the principal pathways of exposure for adults, followed by dust ingestion and air inhalation [21]. 

Notwithstanding the above facts, during the 1980s, TEHP was studied comprehensively for carcinogenicity; however, most of the publicly available in vivo data employed for human health risk assessment are too old [22,23]. In a recent study, TEHP-exposed rats showed testicular pathology and histological changes after 28 days of exposure [24]. The above evidence indicates that TEHP has the ability to accumulate and induce adverse effects to human health. Regardless of that fact, TEHP has not been studied for its hepatotoxic effects in human liver cells. We present the toxicological impact and carcinogenic potential of TEHP in human liver cells (HepG2) owing to its ability for differentiation and genotypic similarities with normal human liver, as well as it is a preferred cell line for the toxicological screening of chemicals [25]. In this study, novel information on TBEP is provided by measuring (i) cytotoxicity, (ii) DNA damage, oxidative stress, mitochondrial dysfunction (Δ*Ψm*), (iii) apoptosis and (iv) transcriptomic alterations in HepG2 cells.

## 2. Materials and Methods

### 2.1. Cell Culture and Cytotoxicity Assays

HepG2 cells were procured from ATCC (Manassas, VA, USA). Experiments were conducted with the 10th passage of cells. HepG2 were seeded and sub-cultured routinely in RPMI-1640 media having 10% FBS and 1% antibiotic solution. Cells were grown in a CO_2_ (5%) incubator at 37 °C (95% humidity). TEHP (CAS #78-42-2, Cat #289922, Sigma Aldrich, St. Louis, MO, USA) exposure concentrations were prepared in DMSO, and prior to exposing the cells it was diluted in serum free medium. To minimize the binding and interference of serum with flame retardants [26,27], we have exposed the HepG2 cells with TEHP in serum free medium (without FBS) for 72 h, and cytotoxicity assessments were carried out by MTT and NRU assays. TEHP screening experiments were carried out at 5, 10, 25, 50, 100, 200, and 400 µM for 24–72 h. No cytotoxicity was found till 50 µM after 72 h (Appendix A). TEHP at the exposure concentrations of 100, 200, and 400 µM showed cytotoxicity after 72 h; hence, these concentrations were selected for further studies. MTT cytotoxicity analysis was performed following the earlier method [28]. Cells were cultured in 96-well culture plate and grown in a CO_2_ incubator. Subsequently, cells were exposed to 100, 200 and 400 µM of TEHP for 72 h. Control cells did not receive any concentration of TEHP. Post exposure, medium was discarded, cells were washed and 10 µL of MTT (5 mg/mL) was added to the wells and incubated for 4 h. Afterwards, medium was removed from the wells, DMSO (200 µL) was added, mixed gently, and the absorbance was read at 550 nm wavelength on a microplate reader (Multiskan Ex, Thermo Fisher Scientific, Vantaa, Finland). Lysosomal toxicity in HepG2 cells was assessed by NRU assay following the previous method [28]. Cells were grown in 96-well culture plate in a CO_2_ incubator. TEHP exposure (100, 200 and 400 µM) was carried out for 72 h. Subsequently, medium was removed, and cells were incubated in 50 µg/mL of neutral red dye for 3 h. Washing was carried out twice with buffer containing 1% CaCl_2_ and 0.5% CH_2_O. Extraction of dye was carried out in 200 µL of dilution buffer (1% acetic acid and 50% ethanol). Absorbance was measured at 550 nm on a microplate reader.

### 2.2. Quantification of ROS, NO, ΔΨm, Ca^++^ Influx, and Esterase Level

Flow cytometric quantification of ROS, NO, Δ*Ψm*, Ca^++^ influx, and esterase level were carried out as described previously [29]. For each of the above parameters HepG2 cells were separately exposed to TEHP (100–400 μM) for 72 h. The treated and control cells were harvested at 3000 rpm for 5 min. Cell pellets were aliquoted in PBS (500 μL). For ROS generation, HepG2 cells were stained with DCFH-DA dye (5 μM); NO staining was carried out with DAF2-DA dye (5 μM); Δ*Ψm* staining was carried out with Rh123 dye (5 μg/mL); Ca^++^ quantification was carried out with 4 µM of Fluo-3 dye; and esterase level in cells were measured with 5 µM of CFDA-SE dye. Staining with above dyes was carried out for at least 1 h in dark at 37 °C in a CO_2_ (5%) incubator. Fluorescence intensity of dyes were recorded in 10,000 cells on a flow cytometer at FL1 channel (Coulter Epics XL/Xl-MCL, Brea, CA, USA).

### 2.3. DNA Damage Analysis

DNA strand breaks in TEHP-treated cells were measured by our previously described method [30]. Control and TEHP (100–400 μM, 72 h)-treated cells were trypsinized and washed twice with PBS. Pellets were resuspended in 100 µL of PBS to which 100 µL of low melting agarose (LMA) was added and mixed gently. A volume of 75 μL was quickly overlayed on a 1/3rd frosted slides and a cover glass was placed to form a uniform layer of gel. Slides were kept on ice to solidify the gel, and 90 μL of LMA was overlayed on each slide. All slides were kept in lysis buffer, electrophoresed, and DNA damage was quantitated by comet assay IV software (Instem, Perceptive Instruments, Cambridge, UK) by following the previous method [30].

### 2.4. Cell Cycle Dysfunction and Apoptosis Analysis

HepG2 cells were exposed to 100–400 µM of TEHP for 72 h in a CO_2_ incubator (5%, 95% humidity). Control cells were run parallel under similar conditions. Post exposure, cells were harvested and washed with PBS. Cells were fixed for 1 h in 500 µL cold ethanol (70%). Cells were washed with PBS, and finally resuspended in 500 µL of PBS containing propidium iodide dye (50 µg/mL). Cell cycle phases of 10,000 cells were recorded on FL3 Log channel of flow cytometer [29]. Apoptosis analysis was carried out by growing the HepG2 cells in the same way as described for the cell cycle analysis. Post exposure time, HepG2 cells were harvested and washed twice with PBS at 1000 rpm for 3 min. Cell pellets were resuspended in 100 µL of binding buffer, to this 10 µL of Annexin V-PE was added, and under dark condition incubated for 15 min on ice. Afterwards, 10 µL of 7-AAD and 380 µL of binding buffer was added. Subsequently, 10,000 cells were analyzed for early, late, and dead cells on a flow cytometer (Muse^®^ Cell Analyzer, Merck Millipore, Burlington, MA, USA).

### 2.5. Apoptotic Protein Analysis

HepG2 cells were exposed to TEHP (100 µM) for 72 h on glass chamber slides in a CO_2_ incubator (5%, 95% humidity). Afterwards, cells were washed in PBS, and fixation was carried out in 70% methanol. Subsequently, blocking was carried out with 5% BSA followed by incubation (1 h) of cells with primary antibodies of P53, caspase-3 and -9 (1:50) (Santa Cruz Biotechnology Inc., Dallas, TX, USA). After incubation time, primary antibodies were washed, and secondary antibodies (Goat anti-rabbit IgG-FITC and goat anti-rabbit IgG-TR (1:100) (Santa Cruz Biotechnology Inc., Dallas, TX, USA) was added and incubated for 1 h. Nuclear staining was carried out with 300 nM of DAPI (4′,6-diamidino-2-phenylindole) for 5 min. After mounting the slides with Prolong Gold Antifade Mountant (Life Technologies, Carlsbad, CA, USA), cell images were grabbed using a fluorescence microscope (Nikon Eclipse 80i, Tokyo, Japan) [29].

### 2.6. Transcriptomic Analysis by RT-qPCR

The transcriptomic analysis has been carried out using RT² Profiler PCR Array, a commercially available 96-well plate using real-time PCR (RT-qPCR) approach. In brief, HepG2 cells were seeded in 6-wells plate and treated with 100 µM of TEHP for 72 h. Total RNA was isolated by using RNeasy Mini Kit (Qiagen, Germantown, MD, USA). Purity of RNA was measured on Nanodrop 8000 and cDNA synthesis was carried out using ready-to-go PCR beads (GE Healthcare, Chalfont Saint Giles, UK). Transcriptomic changes of 84 genes belonging to human cancer pathway was analyzed by RT^2^ Profiler™ PCR Array (PAHS-033Z, SA Biosciences Corporation, Allentown, PA, USA) on Roche^®^ LightCycler^®^ 480 (Roche Diagnostics, Rotkreuz, Switzerland). Expressional changes in TEHP-treated cells were normalized with five housekeeping genes and the results are represented heat map, scatter plot, and fold change [29].

## 3. Results

### 3.1. Cytotoxicity Induced by TEHP

Cytotoxic effects of TEHP in HepG2 cells were quantitated by measuring the activity of mitochondrial dependent MTT dye reduction into formazan product and measuring the lysosomal membrane fragility by NRU assay. Firstly, the morphological analysis of TEHP (100–400 μM, 72 h)-treated cells were examined, which displayed gaps between the adjourning cells and loss of anchorage that led to the appearance of round floating bodies in the culture medium (Figure 1A). On the contrary, control cells retained the morphology of typical epithelial cells. MTT response exhibited a concentration reduction in the survival of exposed cells. TEHP at 100, 200 and 400 μM significantly reduced cell survival to 19.68%, 49.83%, and 58.91% after 72 h of exposure (Figure 1B). Relatively, NRU assay also showed similar effects. HepG2 cells treated with 100, 200 and 400 μM of TEHP for 72 h demonstrated 29.08%, 47.7%, 57.90% reduction in cell survival (Figure 1C).

### 3.2. Intracellular Quantification of ROS, NO, ΔΨm, Ca^++^ Influx, and Esterase 

Flow cytometric analysis was performed to quantitate intracellular ROS, NO, Δ*Ψm*, Ca^++^ influx, and esterase level in TEHP-treated HepG2 cells. In the ROS analysis, HepG2 cells after TEHP (100, 200 μM) treatment showed shifting of MnIX values (13.7% and 12.5%) of DCF towards greater values on the log scale of the flow cytometric graph, as compared to MnIX (7.3%) in the control cells (Figure 2A). HepG2 cells grown in the presence of 100 and 200 μM of TEHP significantly increased the intracellular ROS level to 187.67% and 171.23% (Figure 2B). Moreover, ROS decrement at 400 μM TEHP relates with the higher cell death, indicating the inability of cells to oxidize DCFH-DA. Similarly, intracellular NO analysis exhibited that TEHP (100–400 μM) exposure resulted in the gradational increase in the MnIX values of DAF-2DA recorded as 12.3%, 12.7% and 15.0% (Figure 2C). Cumulative analysis of MnIX data from TEHP (100–400 μM) showed 112.3%, 114.7% and 134.9% greater NO production in cells, as compared to the 100% fluorescence in control (Figure 2D). Quantification of mitochondrial membrane potential (Δ*Ψm*) after TEHP (100, 200 and 400 μM) treatment showed greater MnIX values on the log scale of overlay graph, measured as 10.3%, 12.0%, and 14.9%, indicating higher fluorescence of Rh123 vis à vis control showed MnIX of 8.8% (Figure 2E). Relative to 100% fluorescence in control, TEHP (100-400 μM) exposure resulted in 118.3%, 137.8%, 170.7% higher Rh123 fluorescence indicating the dysfunction of Δ*Ψm* (Figure 2F).

Ca^++^ influx and intracellular esterase level in TEHP-treated HepG2 cells showed fluorescence enhancement of dyes. TEHP (100–400 μM) treated cells exhibited a gradual increase in the MnIX values of Fluo-3 and CFDA-SE dyes, recorded as 16.5%, 17.1%, 19.0% and 9.1%, 10.4%, and 11.0%, while their controls showed MnIX of 10.2% and 7.0%, respectively (Figure 2G,I). Cumulative analysis of Fluo-3 and CFDA-SE MnIX from TEHP (100, 200 and 400 μM)-treated cells showed 150.4%, 162.8%, 180.7% and 129.1%, 149.1%, 160.0% higher Ca^++^ influx and esterase level (Figure 2H,J). Qualitative analysis of Ca^++^ influx also exhibited an increase in the green fluorescence in TEHP (72 h)-treated cells stained with Fluo-3 (Appendix A).

### 3.3. DNA Damage Measurement

Genotoxic effects of TEHP in HepG2 cells were quantitated by measuring the DNA strand breaks using alkaline comet assay. HepG2 cells grown in the presence of TEHP induced significant DNA damage, appearing as a comet tail electrostretched from the nuclear head (Figure 3A). Relative to 0.65 OTM in the control, TEHP (100, 200, and 400 μM)-treated HepG2 cells showed 4.67, 9.35, 13.78-fold greater OTM values (Table 1). Frequency distribution showed the existence of varying levels of DNA damage in HepG2 cells as an effect of TEHP treatments (Figure 3B).

### 3.4. TEHP Trigger Apoptosis in HepG2 Cells

Flow cytometric analysis was carried out to evaluate the disruption cell cycle phases in HepG2 cells after 72 h of TEHP treatment. TEHP-treated cells showed an augmentation in the apoptotic SubG1 peak, depicted in the characteristic cell cycle images (Figure 4A). Cumulative data of SubG1 peak showed 23.1%, 29.6%, and 50.8% of cells in this phase after 72 h of treatment with 100, 200 and 400 μM TEHP; relatively, 9.9% of SubG1 cells were recorded in the control. HepG2 cell cycle phases (G1, S, G2M) were also significantly affected after 72 h of TEHP (100, 200 and 400 μM) treatment (Figure 4B). Overall, TEHP-exposed cells (100, 200 and 400 μM) showed SubG1 arrest as well as a transient G2M arrest in 100 and 200 μM-treated cells. Furthermore, Annexin V-PE staining was carried out to quantify the percentage (%) of cells in early apoptosis, late-apoptosis, and dead phase. The quadrant plots of flow cytometric analysis exhibited apoptosis induction in cells. HepG2 cells grown in the presence of 100, 200 and 400 μM TEHP for 72 h showed 5.26%, 1.00% and 15.67%; 9.58%, 2.49% and 20.57%; 19.80%, 10.25% and 21.55% cells in early apoptotic, late-apoptotic and dead cells. Relatively, control cells showed 3.52%, 0.38% and 4.10% early apoptotic, late-apoptotic and dead cells (Figure 4C). 

### 3.5. Translational Activation of Apoptotic Proteins

Immunofluorescence analysis of DNA damage (P53) and apoptotic proteins (Caspase 3 and 9) in HepG2 cells were analyzed after 72 h treatment with the lowest concentration of TEHP. The fluorescent signal of P53 in HepG2 cells were considerably high in the cytoplasm after TEHP exposure. P53 was conspicuously localized in the nucleolus of TEHP-treated cells. Both caspases (3 and 9) were found localized prominently in the cytoplasm and nucleolus after 72 h of treatment with TEHP (100 μM) (Figure 5).

### 3.6. Activation of Cancer Pathway Genes

The effect of the lowest concentration of TEHP on HepG2 transcriptomic levels were quantitated by using a pathway focused on 84 genes in a 96-well format. TEHP (100 μM)-treated HepG2 cells exhibited upregulation and down regulation of human cancer pathway genes, represented as a heat map and the corresponding list of genes in the array plate (Figure 6A,C). The normalized expression of all genes in the array between TEHP and the control is shown as a scatter plot (Figure 6B). Within the group of over-expressed (>2.0-fold) genes are ATP synthase, H+ transporting, mitochondrial F1 complex, alpha subunit 1, cardiac muscle (*ATP5A1*, 2.63-fold); Growth Arrest and DNA Damage Inducible Gamma (*GADD45G*, 45.13-fold); Glycerol-3-phosphate dehydrogenase 2 (mitochondrial) (*GPD2*, 4.84-fold); Insulin-like growth factor binding protein 5 (*IGFBP5*, 5.19-fold); Antigen identified by monoclonal antibody Ki-67 (*MKI67*, 2.88-fold); Phosphofructokinase, liver (*PFKL*, 9.16-fold); S-phase kinase-associated protein 2 (p45) (*SKP2*, 2.20-fold); Stathmin 1 (*STMN1*, 6.26-fold); Telomeric repeat binding factor 2, interacting protein (*TERF2IP*, 2.12-fold); and Tankyrase, TRF1-interacting ankyrin-related ADP-ribose polymerase 2 (*TNKS2*, 3.15-fold). However, among the under-expressed genes are Erythropoietin (*EPO*, −3.30-fold); Excision repair cross-complementing rodent repair deficiency, complementation group 5 (*ERCC5*, −2.47-fold); Glucose-6-phosphate dehydrogenase (*G6PD*, −2.01-fold); and Mitogen-activated protein kinase 14 (*MAPK14*, −5.26-fold).

## 4. Discussion

Recent evidence has confirmed the presence of TEHP in the blood and urine of elderly people via inhalation of indoor dust. Higher consumption of OPFRs containing food has also been attributed as a prominent a route of TEHP exposure in them [17]. Moreover, an ample amount of TEHP has been reported in COVID-19 face masks, and it has been categorized as a potential substance for human health risk [31]. Given the limited number of studies on the toxicity and carcinogenicity of TEHP, it is imperative to mention that studies on TEHP-induced geno-and-hepatotoxicity are still obscure. Notwithstanding these facts, human hepatocytes or primary hepatocytes (PHHs) are considered as the gold standard when evaluating hepatotoxicity and enzymatic activity, as they preserve ample characteristics of their in vivo CYP450 metabolism after isolation [32]. Nonetheless, there are a few limitations on the frequent application of PHHs, including high variability between donors, limited availability or scarcity of tissue samples, and high expenses owing to their inability to proliferate in vitro [33]. Additionally, PHHs lose their phenotype after a couple of days of culturing, even though the optimized cryopreservation techniques for PHHs have failed in reducing their plateability and CYP450 activity. PHHs accessibility is reliant on organ donation, which is infrequent, and isolation is performed only on marginal livers rejected for transplantation [33,34]. To overcome a few of the above problems, immortalized hepatoma cell lines of human origin have been developed. In this connection, HepG2 cells have been regarded as the most widely used cell line in toxicological research, and are considered an alternative to PHHs [25] as they possess several hepatic functions, ranging from synthesis and secretion of plasma proteins to insulin signaling and bile acid synthesis [35,36,37]. Undoubtedly, the CYP450 (phase I) enzyme expression in HepG2 cells is relatively low compared to PHHs. Nevertheless, expression of some CYPs in HepG2 cells is responsive to some inducers and drugs, which leads to its wider application in the drug metabolism and chemical toxicological research [38,39,40,41]. Moreover, phase II coding genes in HepG2 cells are similar to PHHs; thereby, HepG2 cells might provide protection against promutagens [42]. HepG2 cells, due to their greater differentiation potential and genotypic resemblance to human liver cells, are one of the favored cell lines for analyzing the toxicity of chemicals [25]. Consequently, in this study we have selectively used HepG2 cells for the evaluation of TEHP hepatotoxicity. 

In addition to the survival responses, morphological changes in cells are also a prominent feature to analyze cytotoxicity [43]. Viewing the cobblestone morphology of control cells, TEHP exposure resulted in cell shrinkage, leading to the formation of round floating bodies, specifying cell death. Indistinguishable changes have been reported previously in TEHP-exposed NCI-H1975 cells, indicating apoptotic cell death [15]. We have found that TEHP (100–400 μM) exposure for 72 h significantly declined the survival of HepG2 in MTT and NRU assays, although, no cytotoxicity was observed at lower concentrations (5–50 μM) and early time points (24–48 h). Analogous to our data, NCI-H1975 cells also demonstrated cytotoxic effects at higher concentrations (TEHP, 0–200 μM) at a late time period of 72 h [15]. Taken together, MTT and NRU responses contended a possible interference in the functionality of mitochondria, as well as disturbance in the lysosome membrane in TEHP-exposed HepG2 cells [28]. Hence, we further examined the impact of TEHP on mitochondrial membrane potential (Δ*Ψm*). Flow cytometry data showed a gradual increase in Rh123 fluorescence in the TEHP-exposed cells, which could be related with the mitochondrial intrinsic property to swell, especially under the unfavorable situations such as influx of Ca^++^ and apoptotic cell death [44]. In this line, TEHP-exposed cells also showed significant levels of Ca^++^ influx, affirming its role in mitochondrial dysfunction. We have also found the gradational increase in ROS and NO in TEHP-exposed cells. Mitochondria is a major hub for the production of ROS in the respiratory chain, which disrupts mitochondrial homeostasis; thereby, apoptosis and proapoptotic signals were activated to induce cell death [45]. Consequently, our flow cytometric data affirm the onset of oxidative stress, which affected the mitochondrial functionality of HepG2. In addition to that, TEHP treatment elevated intracellular esterase activity, which has been reported to occur due to obtrusion of cell division [46]; a similar effect has been verified in earlier studies on OPFRs-exposed cells [29,47]. 

Following this, we quantified DNA damage in TEHP-treated cells; a valid reason for such analysis resides in the fact that the above complications act as multifactorial precursors to trigger apoptosis [48]. Comet assay data demonstrated a discernible increase in the broken DNA liberated from TEHP-treated cells, verifying the malfunctioning of the DNA repair system. This is obvious by G2/M arrest, as evident by the greater percentage of cells in the G2/M phase (100 and 200 μM) to fix the damage. However, the transient G2/M arrest was not a permanent and the cells arrested in the SubG1 apoptotic phase (100, 200, 400 μM), clearly indicating a solid reason to recognize the unsuccessful repair of DNA [49]. Similar to our data, TCP-, TECP-, and TDCPP-exposed cell lines have shown DNA damage [29,47,50]. We further quantified the cell cycle phases and verified the activation of apoptotic proteins. The flow cytometric data exhibited significant levels of cell death in TEHP-treated cells, as evident by the greater percentage of cells in the SubG1 apoptotic phase. A similar trend of apoptosis induction was also evidenced in annexin V-PE analysis, validating a substantial number of early apoptotic, late-apoptotic and dead cells after TEHP exposure. Our apoptosis data corroborates well with flow cytometric measurements of apoptosis in TEHP-treated NCI-H1975 cells after 72 h of exposure [15]. Immunolocalization of DNA damage and apoptotic proteins (P53, caspase 3 and 9) in TEHP-treated cells affirmed the onset of mitochondrial depended apoptosis. P53, being a DNA damage response protein, translocates to the nucleus and is known to activate proapoptotic and cell cycle checkpoint genes [51,52], while the initiator caspase 9 becomes activated upon mitochondrial injury and cleaves caspase 3 to effectuate the apoptotic process in cells [53]. Similar to our finding, TCP-, TCEP-, and TDCPP-treated HepG2, RAW264.7 macrophages, and HCECs cells also resulted in the activation of P53, caspase 9 and 3 proteins to effectuate the apoptotic process [29,47,50]. 

Viewing the above cellular anomalies, we were encouraged to pursue the activation of an array of genes belonging to a human cancer pathway and delineate molecular cognizance about the carcinogenic potential of TEHP. In this line, *GADD45A* showed maximal upregulation in the TEHP-treated cells. *GADD45A* is a landmark gene which can sense both non-genotoxic and genotoxic stimulus, including DNA repair, DNA damage and apoptosis [54]. Hence, the activation of *GADD45A* endorses the validity of our data on DNA damage and apoptosis in HepG2 cells. Distantly, *ERCC5* was downregulated in TEHP-treated cells. *ERCC5* is a fundamental gene within the cohort of nucleotide excision repair pathway [55]. Hence, its under-expression affirmed the abrogation of DNA repair machinery in HepG2 cells, which led to DNA damage and apoptosis. On the other hand, *ATP5A1* was upregulated and its activation candidly linked to mitochondrial ATP synthase and catalysis of ATP synthesis. Consequently, the mitochondrial dysfunction in TEHP-treated cells certifies the reliability that the functioning of oxidative phosphorylation may have been disrupted, which may lead to mitochondrial encephalopathy [56]. *GPD2* was also upregulated in TEHP-treated HepG2 cells. *GPD2* governs the catalysis of glycerol-3-phosphate to dihydroxyacetone phosphate in the inner mitochondrial membrane; moreover, it also takes part in the hyper-reduction in ubiquinone and mitochondrial release of ROS [57]. Essentially, the response of mitochondrial dysfunction in TEHP-treated HepG2 cells may have been supplemented by the additional role of *GPD2*. TEHP exposure resulted in the upregulation of *IGFBP5*, which is prominently involved in the cell metabolism, its growth, migration, differentiation and invasion [58,59,60]. Upregulation of *IGFBP5* has been documented in mammary epithelial cells that exhibit DNA damage and apoptosis [61,62]. Hence, in this relationship *IGFBP5* upregulation corresponds with our data on DNA damage, apoptosis and inhibition of cell division (measured in terms of esterase level). *STMN1* was upregulated in TEHP-treated cells. *STMN1* is a hallmark gene for cancer prediction, mainly due to its active involvement in cell proliferation, cell cycle and cancer progression. Thus, its upregulation is related with the cell cycle disruptions and apoptosis induction in TEHP-exposed cells. We also found that *PFKL* was one of the most upregulated genes in the current analysis. *PFKL* plays a fundamental role in glycolysis; its upregulation acts as a turning point in quickly multiplying cancer cells and, most often, its activation fosters proliferation and metastasis via the Warburg effect [63]. Consequently, *PFKL* upregulation entails the existence of imperfect metabolism in HepG2 cells grown in the presence of TEHP. *MAPK14* expression was downregulated, indicating unsuccess in rescuing HepG2 from TEHP-induced apoptosis. Being an important member of the p38 MAPK family, *MAPK14* has been attributed to play a dual role (activator and suppressor) in a variety of cancers by affecting the expression of downstream genes, especially by phosphorylation of other MAPK members [64,65], which need a detailed analysis from the perspective of OPFRs toxicity. Overall, the above data unequivocally affirm the toxicological effects of TEHP in HepG2 cells, especially at higher concentrations, indicating its hepatotoxic potential. It can be surmised that the prolonged exposure of TEHP at high concentrations surpasses the cellular threshold to fix anomalies and damages, eventually triggering apoptotic cell death. Our finding is in close agreement with previous reports on OPFRs, which also exhibited similar toxic effects at high concentrations after extended exposure time periods [66,67].

## 5. Conclusions

MTT and NRU assay responses have affirmed that TEHP disturbs the performance of mitochondrial dehydrogenase, and also affects the membrane integrity of lysosomes in HepG2 cells. TEHP exposure resulted in oxidative stress by the accumulation of ROS and NO; such oxidative stressors affected the mitochondrial potential (Δ*Ψm*), eventually causing Ca^++^ influx. TEHP possess the capability to interact with DNA to induce strand breaks, indicating its genotoxic potential. An advancement in the esterase level specifies the prospect that TEHP may also act as an antiproliferative agent in HepG2 cells. Above anomalies explicitly reveal the reality that TEHP not only affects the structural features of cells, but the functionality of cells is severely compromised, which is evident by the onset of cell death via the mitochondrial-dependent pathway. TEHP-treated cells demonstrated the activation of several genes from a human cancer pathway, indicating its carcinogenic potential. Nonetheless, in-depth studies are warranted using suitable tests models to unravel its cancer-causing mechanism. In total, we provide novel data that TEHP carry credible evidence to act as a hepatotoxic, genotoxic, and a putative carcinogenic agent; hence, we strongly encourage human biomonitoring studies to investigate the link between TEHP exposure, hepatotoxicity and carcinogenicity.

## Figures and Tables

**Figure 1 ijms-23-03998-f001:**
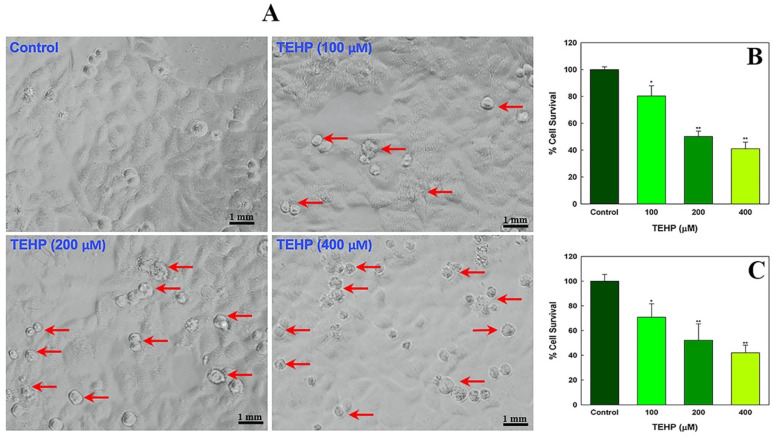
Cytotoxic effects of TEHP in HepG2 cells. (**A**) TEHP-induced structural changes in HepG2 cells after 72 h. Photomicrographs were taken at 20×. (**B**) Mitochondrial dehydrogenase-based cytotoxicity quantification by MTT assay, and (**C**) lysosomal membrane fragility-based NRU assay, indicating lysosomal toxicity. Morphological changes, dead cells are indicated with arrows. * *p* < 0.05, ** *p* < 0.01 versus control.

**Figure 2 ijms-23-03998-f002:**
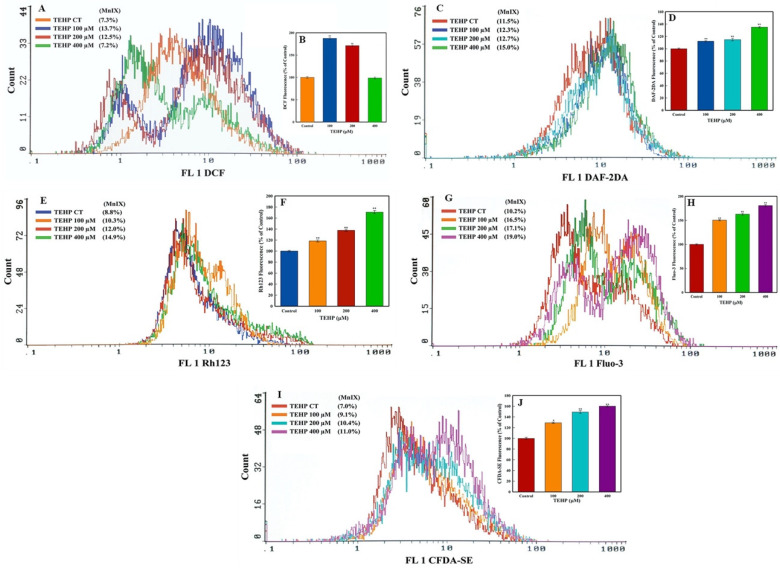
Flow cytometric quantification on intracellular (**A**) ROS, (**C**) NO, (**E**) Δ*Ψm*, (**G**) Ca^++^ influx, and (**I**) esterase levels in HepG2 cells after 72 h of exposure at different concentrations of TEHP. MnIX: mean intensity of dye on the *X*-axis. Histograms in the subfigures (**B**,**D**,**F**,**H**,**J**) are plotted from the mean ± SD of MnIX obtained from three independent experiments carried out in triplicate wells (* *p* < 0.05, ** *p* < 0.01 versus control).

**Figure 3 ijms-23-03998-f003:**
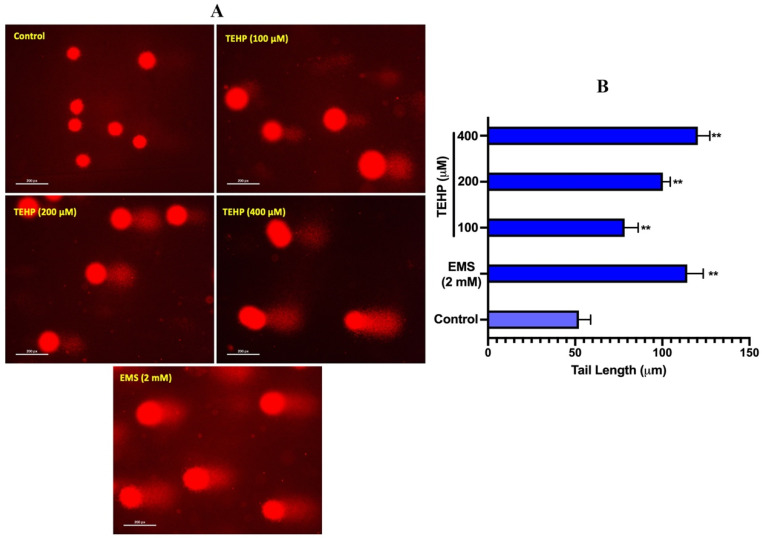
Alkaline comet assay depicting DNA damage in TEHP-treated HepG2 cells. (**A**) Typical comet assay images showing broken DNA liberated as tail from the nuclear head of HepG2 cells after 72 h of TEHP exposure. Ethyl methane sulfonate (EMS) was run as positive control. (**B**) Varying level of comet tail length measured after TEHP exposure. (** *p* < 0.01 versus control).

**Figure 4 ijms-23-03998-f004:**
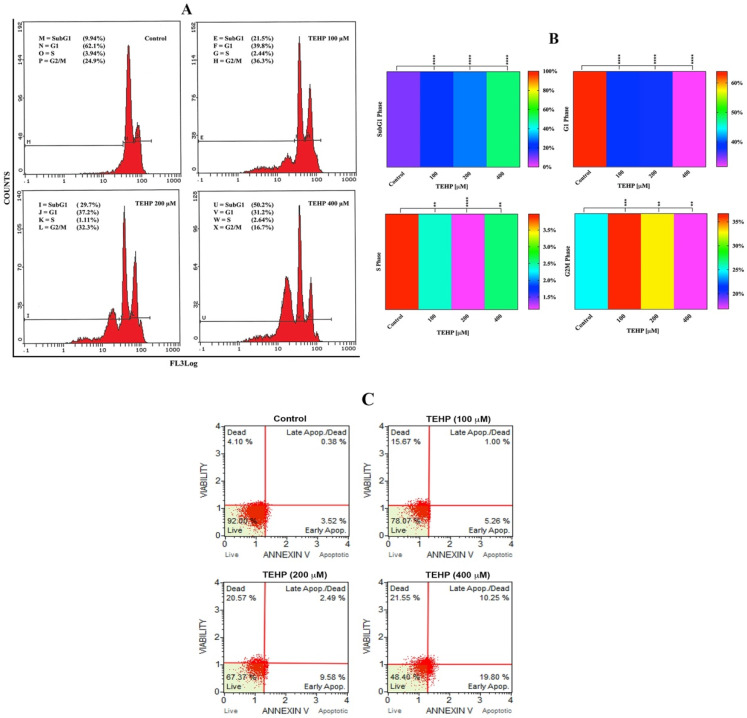
Flow cytometric quantification of cell cycle phases in HepG2 cells exposed to TEHP for 72 h. (**A**) Representative cell cycle images from TEHP-treated HepG2 cells showing a concentration dependent increase in the apoptotic peak (SubG1). (**B**) Cumulative data of all phases are depicted as heat map derived from three independent experiments carried out in triplicate wells. SubG1, G1 and S phases (***** p <* 0.0001); S phase (** *p* 0.0017 Control vs. 100, ** *p* 0.0038 Control vs. 400); G2M phase (*** *p* 0.0006 Control vs. 100, ** *p* 0.0067 Control vs. 200, ** *p* 0.0072 Control vs. 400). (**C**) Early apoptotic, late-apoptotic and dead cells quantification in control and TEHP-treated cells after 72 h.

**Figure 5 ijms-23-03998-f005:**
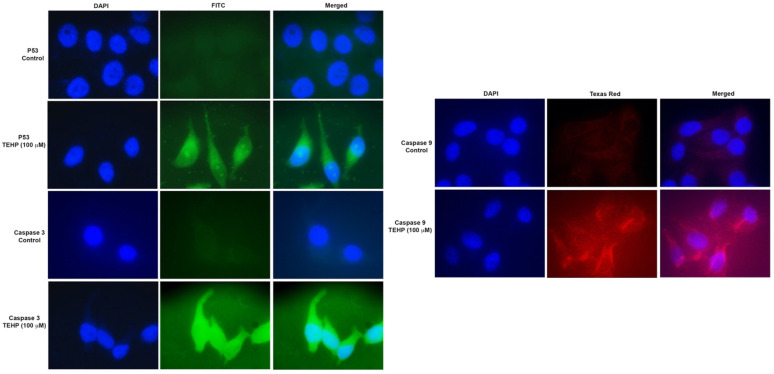
Translational activation of apoptotic proteins (P53 and caspases 3 and 9) in TEHP (100 μM)-treated (72 h) HepG2 cells. Images were captured at 100× magnification by the use of a fluorescence microscope.

**Figure 6 ijms-23-03998-f006:**
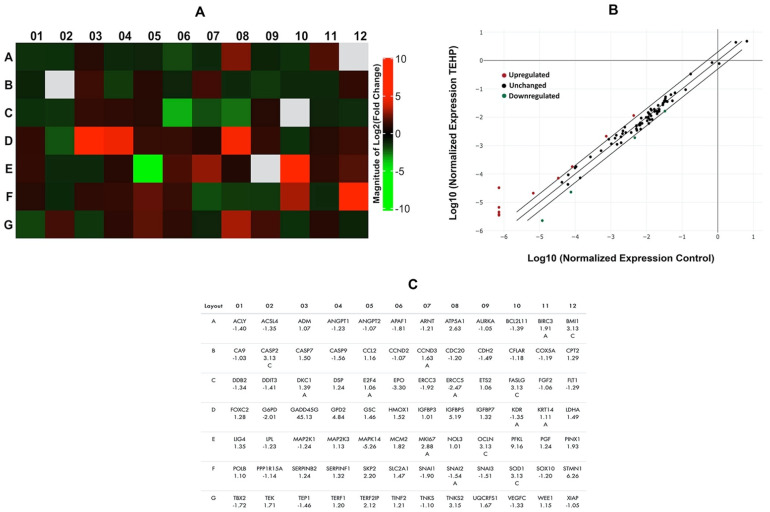
Transcriptomic changes in HepG2 cells after 72 h of TEHP (100 μM) exposure. (**A**) Heat map displaying upregulated (red boxes), downregulated (green boxes), and (gray boxes) undetermined genes in the array. (**B**) Genes that has exceeded the threshold of >2.0-fold regulation is shown in scatter plot. (**C**) Annotation of human cancer pathway genes, their location in array plate and fold regulation is depicted in the panel.

**Table 1 ijms-23-03998-t001:** TEHP-induced DNA damage in HepG2 cells, analyzed using different parameters of alkaline comet assay.

Groups	Olive Tail Moment (Arbitrary Unit)	Tail Intensity (%)
Control	0.65 ± 0.09	3.23 ± 0.33
EMS (2 mM)	12.26 ± 1.88 **	34.21 ± 2.65 **
**TEHP (µM)**		
100	3.04 ± 0.41 **	10.54 ± 1.39 **
200	6.08 ± 1.40 **	14.27 ± 0.92 **
400	8.96 ± 0.92 **	20.23 ± 1.66 **

Data are the mean ± S.D. of three independent experiments carried out in triplicate slides. ** *p* < 0.01; EMS: Ethyl methanesulphonate.

## Data Availability

Not applicable.

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
