# Peer review of "Cyto-Genotoxic and Transcriptomic Alterations in Human Liver Cells by Tris (2-Ethylhexyl) Phosphate (TEHP): A Putative Hepatocarcinogen"

_ijms, 2022, doi:10.3390/ijms23073998_

Round 1

Reviewer 1 Report

The author would like to mention about cellular genotoxicity of a test substance THEP on human hepatocytes. The experimental design and logic what author insist were clear, but there are several things to think about.

Major consideration of this article

  1.  Can author insist that it is the effect of human hepatocytes only using HepG2 cell line. Because HepG2 cell was immortalized cell line, therefore author has to be use primary human hepatocytes to identify the effect of THEP.
  2. There are only late time point-high dose result and mention about early time point-low dose had no cytotoxic effect. For the argument to be supported, it has to be show late time point-low dose results for example 72h exposured under 50uM THEP treated groups.
  3. Using flow-cytometry Ca influx measurement was not precise method, therefore author better to show the data of ca influx by measure ca imaging of THEP exposured HepG2 cell.
  4. In the transcriptomic analysis, It would be good to clearly mention what author want to talk about and the method of RNA sequencing.

Minor consideration 

  1. In the Figure2, If author obtained 3 independent experiment and each triplicates the graph would be better to show statistical differences.
  2. And also in Figure2, it seems that you can better show what you want to claim through smoothness control.
  3. In figure2A,B, It need to explain about decrement of ROS in  high dose(400uM) of THEP.

Author Response

Reviewer #1

The author would like to mention about cellular genotoxicity of a test substance THEP on human hepatocytes. The experimental design and logic what author insist were clear, but there are several things to think about.

Major Comment 1

Can author insist that it is the effect of human hepatocytes only using HepG2 cell line. Because HepG2 cell was immortalized cell line, therefore author has to be use primary human hepatocytes to identify the effect of THEP.

Author Response Major Comment 1

We appreciate this pertinent query raised by our learned reviewer. Human hepatocytes or primary hepatocytes (PHHs) are considered as the gold standard to evaluate the CYP3A4 activity and regulation, as they preserve ample characteristics of their in vivo CYP450 metabolism after the isolation [1]. In spite of the fact, there are few restraints on the frequent application of PHHs including high variability between donors, limited availability or per se scarcity of tissue samples, and expensive owing to their inability to proliferate in vitro [2]. In addition, PHHs lose their phenotype after a couple of days of culturing. Even though the optimized cryopreservation techniques for PHHs have failed in reduce their plateability and CYP450 activity. Also, PHHs accessibility is reliant on organ donation, which is infrequent, and the isolation is performed only on marginal livers rejected for transplantation [2,3].  

To overcome few of the above problems, immortalized hepatoma cell lines from human origin have been developed. In this connection, HepG2 cells have been regarded as most widely used cell line in toxicological research, and seen alternative to PHHs [4]. As it possesses several hepatic functions ranging from synthesis and secretion of plasma proteins, insulin signaling, and bile acid synthesis [5-7]. Undoubtedly, the CYP450 (phase I) enzyme expression in HepG2 cells is relatively low as compared to PHHs. Nevertheless, expression of some CYPs in HepG2 cells is responsive to some inducers and drugs, which leads to its wider application in the drug metabolism and chemical toxicological research [8-11]. Moreover, the phase II coding genes in HepG2 cells are similar to PHHs; thereby, HepG2 cells might provide protection against promutagens [12].

We agree with the reviewer’s remark that TEHP toxicity should be tested in PHHs; however, in this study, we primarily focused to evaluate the genotoxicity and cytotoxicity induced by TEHP, and not to go deep-down on the biotransformation of TEHP. Moreover, in our current facility we do not have resources to get human tissues samples and isolate PHHs. Hence, we have selectively chosen HepG2 cells to evaluate the hepatotoxicity of TEHP. Based on our toxicity data, we hypothesized that TEHP may be activated by some human CYPs enzymes to induce DNA damage, which is a mechanism for chemical carcinogenesis, and warrants future studies involving human PHHs.

For better clarity to the readers, we have incorporated the above justification in the revised manuscript. Please see lines 332-355.

Major Comment 2

There are only late time point-high dose result and mention about early time point-low dose had no cytotoxic effect. For the argument to be supported, it has to be show late time point-low dose results for example 72h exposured under 50uM THEP treated groups.

Author Response Major Comment 2

Reviewer has raised a valid point and we appreciate it. Hence, in this regard we have performed new experiment by exposing the HepG2 cells with low concentrations of TEHP (0.5, 1, 2, 5, 10, 25, 50 μM for 72 and measured its cytotoxicity by MTT assay. The data showed no cytotoxic effects of TEHP at the above concentrations after 72 h of exposure. Above response is now added in the revised manuscript as supplementary data file. Please see Supplementary Figures S1 and S2 and lines 102-103 in the revised manuscript.

Major Comment 3

Using flow-cytometry Ca influx measurement was not precise method, therefore author better to show the data of ca influx by measure ca imaging of THEP exposured HepG2 cell.

Author Response Major Comment 3

As suggested, we have performed a new experiment for the qualitative analysis of Ca++ influx in HepG2 cells after TEHP exposure for 72h. Results are provided as Supplementary Figure S3 and also described in lines 229-231 of the revised manuscript.

Major Comment 4

In the transcriptomic analysis, It would be good to clearly mention what author want to talk about and the method of RNA sequencing.

Author Response Major Comment 4

Reviewer remark is greatly appreciated. The transcriptomic analysis in our study has been done using RT² Profiler PCR Array, a commercially available 96-well plate, for quantifying the expression of a focused panel of genes belonging to human cancer pathways. This was done using real-time PCR (RT-qPCR) approach. These information’s are now incorporated in the revised manuscript. Please see lines 172-173.

Minor Comment 1

In the Figure2, If author obtained 3 independent experiment and each triplicates the graph would be better to show statistical differences.

Author Response Minor Comment 1

Histograms shown as inset in each panel has been derived from the experimental repeats and plotted as mean ± S.D. values, and the statistical difference between the treatment groups are shown as asterisks (*p<0.05,  **p<0.01) versus control, as also shown in figure legend.  

No changes have been made.

Minor Comment 2

And also in Figure2, it seems that you can better show what you want to claim through smoothness control.

Author Response Minor Comment 2

We appreciate reviewer’s concern. However, our flow cytometer and its software are quite old, and do not have option for smoothness control, so we are unable to do so and present the data as such generated by the machine.

No changes have been made.

Minor Comment 3

In figure2A,B, It need to explain about decrement of ROS in  high dose(400uM) of THEP.

Author Response Minor Comment 2

ROS decrement at 400 μM TEHP relates with the higher cell death, indicating the inability of cells to oxidize DCFH-DA. This justification in now added in the revised manuscript. Please see lines 211-212.

Note: The references cited above in response to reviewer 1 are already incorporated in the revised manuscript.

  1. LeCluyse, E.L. Human hepatocyte culture systems for the in vitro evaluation of cytochrome P450 expression and regulation. Eur. J. Pharm. Sci. 2001, 13, 343-368, doi:https://doi.org/10.1016/S0928-0987(01)00135-X.
  2. Li, A.P. Human hepatocytes: Isolation, cryopreservation and applications in drug development. Chemico-Biological Interactions 2007, 168, 16-29, doi:https://doi.org/10.1016/j.cbi.2007.01.001.
  3. Roymans, D.; Van Looveren, C.; Leone, A.; Parker, J.B.; McMillian, M.; Johnson, M.D.; Koganti, A.; Gilissen, R.; Silber, P.; Mannens, G., et al. Determination of cytochrome P450 1A2 and cytochrome P450 3A4 induction in cryopreserved human hepatocytes. Biochemical Pharmacology 2004, 67, 427-437, doi:https://doi.org/10.1016/j.bcp.2003.09.022.
  4. Gerets, H.; Tilmant, K.; Gerin, B.; Chanteux, H.; Depelchin, B.; Dhalluin, S.; Atienzar, F. Characterization of primary human hepatocytes, HepG2 cells, and HepaRG cells at the mRNA level and CYP activity in response to inducers and their predictivity for the detection of human hepatotoxins. Cell biology and toxicology 2012, 28, 69-87.
  5. Javitt, N.B. Hep G2 cells as a resource for metabolic studies: lipoprotein, cholesterol, and bile acids. The FASEB Journal 1990, 4, 161-168.
  6. Dongiovanni, P.; Valenti, L.; Fracanzani, A.L.; Gatti, S.; Cairo, G.; Fargion, S. Iron depletion by deferoxamine up-regulates glucose uptake and insulin signaling in hepatoma cells and in rat liver. The American journal of pathology 2008, 172, 738-747.
  7. Donato, M.T.; Tolosa, L.; Gómez-Lechón, M.J. Culture and Functional Characterization of Human Hepatoma HepG2 Cells. In Protocols in In Vitro Hepatocyte Research, Vinken, M., Rogiers, V., Eds. Springer New York: New York, NY, 2015; 10.1007/978-1-4939-2074-7_5pp. 77-93.
  8. Gomez-Lechon, J.M.; Donato, T.M.; Lahoz, A.; Castell, V.J. Cell Lines: A Tool for In Vitro Drug Metabolism Studies. Curr. Drug Metab. 2008, 9, 1-11, doi:http://dx.doi.org/10.2174/138920008783331086.
  9. Liu, C.-L.; Lim, Y.-P.; Hu, M.-L. Fucoxanthin attenuates rifampin-induced cytochrome P450 3A4 (CYP3A4) and multiple drug resistance 1 (MDR1) gene expression through pregnane X receptor (PXR)-mediated pathways in human hepatoma HepG2 and colon adenocarcinoma LS174T cells. Mar. Drugs 2012, 10, 242-257.
  10. Tirona, R.G.; Lee, W.; Leake, B.F.; Lan, L.-B.; Cline, C.B.; Lamba, V.; Parviz, F.; Duncan, S.A.; Inoue, Y.; Gonzalez, F.J., et al. The orphan nuclear receptor HNF4α determines PXR- and CAR-mediated xenobiotic induction of CYP3A4. Nat. Med. 2003, 9, 220-224, doi:10.1038/nm815.
  11. Chen, Z.; Xie, J.; Li, Q.; Hu, K.; Yang, Z.; Yu, H.; Liu, Y. Human CYP enzyme-activated clastogenicity of 2-ethylhexyl diphenyl phosphate (a flame retardant) in mammalian cells. Environmental Pollution 2021, 285, 117527.
  12. Wilkening, S.; Stahl, F.; Bader, A. Comparison of primary human hepatocytes and hepatoma cell line Hepg2 with regard to their biotransformation properties. Drug Metab. Disposition 2003, 31, 1035-1042.

Reviewer 2 Report

Review comments

The review paper by Saquib et al on “Cyto-Genotoxic and Transcriptomic Alterations in Human Liver Cells by Tris (2-ethylhexyl) phosphate (TEHP): A Putative Hepatocarcinogen” is an interesting research study. However, here are some of my major concerns which should be corrected:

  1. Although many studies use the hepG2 cell line for evaluation of cytotoxicity. However, it is still a hepatocellular carcinoma cell line having a chromosome number 55, Which means they cannot be treated as a normal cell. Please explain the rationale in using this cell line and you should add one more liver cell line which is not cancerous.
  2. The study will be even better if the authors use a normal hepatic cell line and provide Primary hepatocytes isolated from the liver are effective tools for the in vitroevaluation of metabolism, drug-drug interactions, hepatotoxicity, and transporter activity. It can be purchased from ATCC as well.
  3. The authors should explain in detail the consequences of exposure to high concentration of organophosphate flame retardants with special emphasis on Tris(2-ethylhexyl) phosphate (TEHP)
  4. The authors should perform Annexin V Flow cytometry for detection of apoptosis in HepG2 cells treated with different concentration of
  5. The authors need to expand the result sections and elaborate more and explain with detail and references included for their findings in comparison with similiar studies.
  6. Please explain at what stage of cell cycle were the cells arrested? Was it Sub G1, G1, S, G2M).
  7. The authors should briefly in a short way explain the activation of p53, caspase 3 and caspase 9 in apoptosis.
  8. In the result section 3.1, the authors have mentioned that 100 μM of TEHP is causing 12±0.92% of cell death. However, in Figure 6, we see a high expression of p53, caspase 3 and caspase 9. How do the authors justify the obtained results?
  9. In qPCR array of 84 genes, HepG2 cells treated with TEHP where the cancer pathway genes were overexpressed. The authors should briefly explain overexpression of ki67 gene which encodes a nuclear protein that is associated cellular proliferation. As well as the downregulated expression of Mitogen-activated protein kinase 14 (MAPK14), which is also involved in a wide variety of cellular processes such as proliferation, differentiation, transcription regulation and development.
  10. I found this study and manuscript to be strikingly similar with the previous published papers from the same authors. Please justify, so that there is no duplication of images after publication for the safety of the concerned authors and the journal.

Saquib, Q.; Al-Salem, A.M.; Siddiqui, M.A.; Ansari, S.M.; Zhang, X.; Al-Khedhairy, A.A. Organophosphorus Flame Retardant TDCPP Displays Genotoxic and

Carcinogenic Risks in Human Liver Cells. Cells 2022, 11, 195. https://doi.org/10.3390/cells11020195

Minor comment

  1. How did the authors forget to add line numbers. How can we the reviewers point the mistakes in the manuscript for easy evaluation and correction
  2. In the abstract. Herein, we find that HepG2 cells exposed to TEHP (100, 200, 400 μM) for 72h “declined” the cell survival to 19.68%. Please change the word to “Minimized” or “reduced”.
  3. Elderly persons exposed to TEHP via inhalation of indoor dust were frequently detected with higher “concertation” in blood. Please change to concentration.
  4. How were the cell line HepG2 acquired? Please add the details for the purchase of hepG2 from which vendor and also write down the passage number before the authors started the experiment.
  5. “Human liver cells (HepG2) were grown in RPMI-1640 media (without FBS) containing 1% antibiotic solution”. As per the ATCC guidelines for culturing HepG2 cells, it mentiones “The base medium for this cell line is ATCC-formulated Eagle's Minimum Essential Medium, Catalog No. 30-2003. To make the complete growth medium, add the following components to the base medium: fetal bovine serum to a final concentration of 10%”. Why was 10% FBS not used, please explain the rationality behind it?
  6. Add a space after every 72h as in 72 h and make necessary changes wherever needed.
  7. Please provide better image quality for all figures, especially figure 2.
  8. Please provide a better statistical graph for Figure 3B.

Author Response

Reviewer #2

The review paper by Saquib et al on “Cyto-Genotoxic and Transcriptomic Alterations in Human Liver Cells by Tris (2-ethylhexyl) phosphate (TEHP): A Putative Hepatocarcinogen” is an interesting research study. However, here are some of my major concerns which should be corrected:

Major Comment 1

Although many studies use the hepG2 cell line for evaluation of cytotoxicity. However, it is still a hepatocellular carcinoma cell line having a chromosome number 55, Which means they cannot be treated as a normal cell. Please explain the rationale in using this cell line and you should add one more liver cell line which is not cancerous.

Author Response Major Comment 1

Reviewer remark is greatly appreciated. Indeed, we agree with the fact that HepG2 cells are not normal cells. In spite of the fact there are several advantages associate in using this cell line for varying types of biological studies. HepG2 cells have been regarded as most widely used cell line in toxicological research, and seen as alternative to primary hepatocytes (PHHs) [4]. As it possesses several hepatic functions ranging from synthesis and secretion of plasma proteins, insulin signaling, and bile acid synthesis [5-7]. Undoubtedly, the CYP450 (phase I) enzyme expression in HepG2 cells is relatively low, as compared to PHHs. Nevertheless, expression of some CYPs in HepG2 cells is responsive to few inducers and drugs, which leads to its larger application in the drug metabolism and chemical toxicological research [8-11]. Moreover, the phase II coding genes in HepG2 cells are similar to PHHs; thereby, HepG2 cells might provide protection against promutagens [12].

At the moment, in our lab we do not have non-cancerous cell line; hence, we are unable to add in our study. After COVID-19 outbreak, procurement of a new cell line is extremely tough and involve substantial clearances from governmental authorities, which we cannot do currently to add in this study. Although, we value reviewer suggestion and try to procure normal cells in our lab for future studies. We hope that our learned reviewer will understand current limitation.

For the better clarity to readers, above justification is now added in the revised manuscript. Please see lines 332-355.

Major Comment 2

The study will be even better if the authors use a normal hepatic cell line and provide Primary hepatocytes isolated from the liver are effective tools for the in vitroevaluation of metabolism, drug-drug interactions, hepatotoxicity, and transporter activity. It can be purchased from ATCC as well.

Author Response Major Comment 2

We appreciate this pertinent query raised by our learned reviewer. Human hepatocytes or primary hepatocytes (PHHs) are considered as the gold standard to evaluate the CYP3A4 activity and regulation, as they preserve ample characteristics of their in vivo CYP450 metabolism after the isolation [1]. In spite of the fact, there are few restraints on the frequent application of PHHs including high variability between donors, limited availability or per se scarcity of tissue samples, and expensive owing to their inability to proliferate in vitro [2]. In addition, PHHs lose their phenotype after a couple of days of culturing. Even though the optimized cryopreservation techniques for PHHs have failed in reduce their plateability and CYP450 activity. Also, PHHs accessibility is reliant on organ donation, which is infrequent, and the isolation is performed only on marginal livers rejected for transplantation [2,3]. 

To overcome few of the above problems, immortalized hepatoma cell lines from human origin have been developed. In this connection, HepG2 cells have been regarded as most widely used cell line in toxicological research, and seen alternative to PHHs [4]. As it possesses several hepatic functions ranging from synthesis and secretion of plasma proteins, insulin signaling, and bile acid synthesis [5-7]. Undoubtedly, the CYP450 (phase I) enzyme expression in HepG2 cells is relatively low as compared to PHHs. Nevertheless, expression of some CYPs in HepG2 cells is responsive to some inducers and drugs, which leads to its wider application in the drug metabolism and chemical toxicological research [8-11]. Moreover, the phase II coding genes in HepG2 cells are similar to PHHs; thereby, HepG2 cells might provide protection against promutagens [12].

We agree with the reviewer’s remark that TEHP toxicity should be tested in PHHs; however, in this study, we primarily focused to evaluate the genotoxicity and cytotoxicity induced by TEHP, and not to go deep-down on the biotransformation of TEHP. Moreover, in our current facility we do not have resources to get human tissues samples and isolate PHHs. Hence, we have selectively chosen HepG2 cells to evaluate the hepatotoxicity of TEHP. Based on our toxicity data, we hypothesized that TEHP may be activated by some human CYPs enzymes to induce DNA damage, which is a mechanism for chemical carcinogenesis, and warrants future studies involving human PHHs.

At the moment, in our lab we do not have non-cancerous cell line; hence, we are unable to add in our study. After COVID-19 outbreak, procurement of a new cell line is extremely tough and involves substantial clearances from governmental authorities. We hope that our learned reviewer will understand such limitation.

For better clarity to the readers, we have incorporated the above justification in the revised manuscript. Please see lines 332-355.

Major Comment 3

The authors should explain in detail the consequences of exposure to high concentration of organophosphate flame retardants with special emphasis on Tris(2-ethylhexyl) phosphate (TEHP)

Author Response Major Comment 3

As suggested, we have added the consequences of OPFRs exposure at higher concentrations, in particular TEHP exposure has also been incorporated. Please see lines 437-447 in the revised manuscript.

Major Comment 4

The authors should perform Annexin V Flow cytometry for detection of apoptosis in HepG2 cells treated with different concentration of

Author Response Major Comment 4

As per the demand of reviewer we have performed a new experiment for TEHP exposed HepG2 cells and measured the early, late and necrotic cell population using Annexin V kit. Please see section 2.4 lines 146-159 for methodology, section 3.4 lines 264-272 for result, revised figure 4 (panel C), and lines 392-394 for discussion in the revised manuscript.

Major Comment 5

The authors need to expand the result sections and elaborate more and explain with detail and references included for their findings in comparison with similar studies.

Author Response Major Comment 5

As indicated results are elaborated and expanded. Please see results section 3.1 (lines 185-187), section 3.2 (lines 205-207, 211-214, 217-219, 223-224), section 3.3. (lines 240-241), section 3.4 (lines 258-259, 264-272), section 3.5 (lines 281-283), section 3.6 (lines 300-305).

Major Comment 6

Please explain at what stage of cell cycle were the cells arrested? Was it Sub G1, G1, S, G2M).

Author Response Major Comment 6

As pointed out by the reviewer, the cells were arrested in G2/M phase. This information is incorporated in the revised manuscript. Please see section 3.4 (lines 266), discussion (lines 385-386).

Major Comment 7

The authors should briefly in a short way explain the activation of p53, caspase 3 and caspase 9 in apoptosis.

Author Response Major Comment 7

As desired, a brief discussion on the explanation on the activation of p53, caspase 3 and caspase 9 in apoptosis is now added in the revised manuscript. Please see lines 397-402.

 Major Comment 8

In the result section 3.1, the authors have mentioned that 100 μM of TEHP is causing 12±0.92% of cell death. However, in Figure 6, we see a high expression of p53, caspase 3 and caspase 9. How do the authors justify the obtained results?

Author Response Major Comment 8

We appreciate reviewer’s remark. However, we would like to draw your kind attention that in section 3.1. we did not mention that 100 μM of TEHP is causing 12±0.92% of cell death. In fact, it is written that 100 μM of TEHP has caused 19.68% cytotoxicity in MTT assay and 29.08% lysosomal toxicity in NRU assay, which was validated as apoptotic cell death in flow cytometric data (cell cycle and annexin V). Hence, higher expression of p53, caspase 3 and caspase 9 are obvious.

No changes have been made.

Major Comment 9

In qPCR array of 84 genes, HepG2 cells treated with TEHP where the cancer pathway genes were overexpressed. The authors should briefly explain overexpression of ki67 gene which encodes a nuclear protein that is associated cellular proliferation. As well as the downregulated expression of Mitogen-activated protein kinase 14 (MAPK14), which is also involved in a wide variety of cellular processes such as proliferation, differentiation, transcription regulation and development.

Author Response Major Comment 9

We appreciate reviewer’s concern. In this study, our prime goal was to focus on the panel of genes belonging to human cancer pathways. The transcriptomic analysis has been done using RT² Profiler PCR Arrays, a commercially available 96-well plate. This assay plate did not contain the ki67 and MAPK14 primers and hence, not included in the panel. However, we value the reviewer’s suggestion, and assure that the indicated genes will be included in our future studies, especially while quantifying the expression of individual genes.

No changes have been made.

Major Comment 10

I found this study and manuscript to be strikingly similar with the previous published papers from the same authors. Please justify, so that there is no duplication of images after publication for the safety of the concerned authors and the journal.

Saquib, Q.; Al-Salem, A.M.; Siddiqui, M.A.; Ansari, S.M.; Zhang, X.; Al-Khedhairy, A.A. Organophosphorus Flame Retardant TDCPP Displays Genotoxic and

Carcinogenic Risks in Human Liver Cells. Cells 2022, 11, 195. https://doi.org/10.3390/cells11020195

Author Response Major Comment 10

We appreciate your concern and hereby present the flowing justification. We have a project on OPFRs induced hepatotoxicity toxicity in HepG2 cells, in which we have studied different OPFRs, including the current one on TEHP. Since, the project objectives were same, similar battery of tests were performed for all OPFRs. Consequently, lots of data were generated, which cannot be summarized in one article; hence, we have decided to report them in separate publications, including the one which you have mentioned. The published articles reported distinctive responses of HepG2 cells upon exposure with different OPFRs and none of the data images are duplicated.

Since the reviewer has a concern, we assure that the none of the data image, including the current study, are duplicated in any of the published papers from our group.

Minor Comment 1

How did the authors forget to add line numbers. How can we the reviewers point the mistakes in the manuscript for easy evaluation and correction

Author Response Minor Comment 1

In the revised manuscript we have added the line numbers for your convenience.

Minor Comment 2

In the abstract. Herein, we find that HepG2 cells exposed to TEHP (100, 200, 400 μM) for 72h “declined” the cell survival to 19.68%. Please change the word to “Minimized” or “reduced”.

Author Response Minor Comment 2

As indicated, we have replaced the word “declined” to “reduced”. Please see abstract (line 19).

Minor Comment 3

Elderly persons exposed to TEHP via inhalation of indoor dust were frequently detected with higher “concertation” in blood. Please change to concentration.

Author Response Minor Comment 3

Our sincere apologies for typos. The word “concertation” is now corrected with correct word “concentration”. Please see line 65.

Minor Comment 4

How were the cell line HepG2 acquired? Please add the details for the purchase of hepG2 from which vendor and also write down the passage number before the authors started the experiment.

Author Response Minor Comment 4

HepG2 was procured from ATCC ((Manassas, VA, USA). Experiments were conducted with the 10th passage of HepG2 cells. This information is added in the revised manuscript. Please see lines 91-92.

Minor Comment 5

“Human liver cells (HepG2) were grown in RPMI-1640 media (without FBS) containing 1% antibiotic solution”. As per the ATCC guidelines for culturing HepG2 cells, it mentiones “The base medium for this cell line is ATCC-formulated Eagle's Minimum Essential Medium, Catalog No. 30-2003. To make the complete growth medium, add the following components to the base medium: fetal bovine serum to a final concentration of 10%”. Why was 10% FBS not used, please explain the rationality behind it?

Author Response Minor Comment 5

Reviewer’s concern is very pertinent and rightly pointed. We regret that the following statement “Human liver cells (HepG2) were grown in RPMI-1640 media (without FBS) containing 1% antibiotic solution” is not delivering a clear meaning. In fact, HepG2 cells used in this study were regularly cultured in the presence of recommended 10% FBS. However, HepG2 cells were treated for 72h with different concentrations of TEHP in serum free medium (without FBS). A proper rationale for such condition resides in the fact that there are reports that flame retardants bind to the serum present in the culture medium. Therefore, to minimize the binding and interference of serum with flame retardants [26,27], we have exposed HepG2 cells with TEHP in serum free medium (without FBS) for 72h. This justification is now added in the revised manuscript. Please see lines 92-98.

Minor Comment 6

Add a space after every 72h as in 72 h and make necessary changes wherever needed.

 Author Response Minor Comment 6

As suggested, 72h is now replaced with 72 h. Please see the revised manuscript and the changes is highlighted.

Minor Comment 7

Please provide better image quality for all figures, especially figure 2.

 Author Response Minor Comment 7

As suggested, quality of all figures (1-7) are now improved (600 dpi) and provided in revised manuscript. Please see figures 1-7.

Minor Comment 8

Please provide a better statistical graph for Figure 3B.

 Author Response Minor Comment 8

As desired, statistical graph for Figure 3B is now incorporated in the revised manuscript.

 Note: The references cited above in response to reviewer 2 are already incorporated in the revised manuscript.

Round 2

Reviewer 2 Report

The authors have clearly answered all my questions in detail. I feel the manuscript is good for acceptance after this minor correction.

Concern 1

In my first review, I have asked this question

Major Comment 6

Please explain at what stage of cell cycle were the cells arrested? Was it Sub G1, G1, S, G2M).

Author Response Major Comment 6

As pointed out by the reviewer, the cells were arrested in G2/M phase. This information is incorporated in the revised manuscript. Please see section 3.4 (lines 266), discussion (lines 385-386).

As per your data, the highest peak is seen in Sub G1 phase at 400 µM treatment group which is in accordance with studies published by previous researches. Mostly Cells undergo cell cycle arrest at sub G1 phase during apoptosis and your Sub G1 cell cycle arrest peak is very high in consistent with the findings from other researchers (reference 1, 2) in figure 4A. Please correct this in your manuscript by adding the peaks was also observed in Sub G1 at 100 and 200 µM.  Also make changes in Lines 385-386.

  1. Wang YY, Kwak JH, Lee KT, Deyou T, Jang YP, Choi JH. Isoflavones isolated from the seeds of Millettia ferruginea induced apoptotic cell death in human ovarian cancer cells. Molecules. Published online 2020. doi:10.3390/molecules25010207
  2. Chatterjee N, Das S, Bose D, Banerjee S, Jha T, Das Saha K. Lipid from infective L. Donovani regulates acute myeloid cell growth via mitochondria dependent MAPK pathway. PLoS One. Published online 2015. doi:10.1371/journal.pone.0120509

Concern 2

In Line 308-309, The authors have shown Mitogen-activated protein kinase 14 (MAPK14, -5.26-fold) increase. As per the publications. MAPK14 main role could be in protecting cells from apoptosis.

  1. Liu J, Yu X, Yu H, Liu B, Zhang Z, Kong C, Li Z. Knockdown of MAPK14 inhibits the proliferation and migration of clear cell renal cell carcinoma by downregulating the expression of CDC25B. Cancer Med. Published online 2020. doi:10.1002/cam4.2795
  2. Fang JY, Richardson BC. The MAPK signalling pathways and colorectal cancer. Lancet Oncol. Published online 2005. doi:10.1016/S1470-2045(05)70168-6

Please make corrections. I have no other questions. Accept it after this minor correction.

Author Response

Reviewer #2

The authors have clearly answered all my questions in detail. I feel the manuscript is good for acceptance after this minor correction.

Concern 1

In my first review, I have asked this question

Major Comment 6

Please explain at what stage of cell cycle were the cells arrested? Was it Sub G1, G1, S, G2M).

Author Response Major Comment 6

As pointed out by the reviewer, the cells were arrested in G2/M phase. This information is incorporated in the revised manuscript. Please see section 3.4 (lines 266), discussion (lines 385-386).

As per your data, the highest peak is seen in Sub G1 phase at 400 µM treatment group which is in accordance with studies published by previous researches. Mostly Cells undergo cell cycle arrest at sub G1 phase during apoptosis and your Sub G1 cell cycle arrest peak is very high in consistent with the findings from other researchers (reference 1, 2) in figure 4A. Please correct this in your manuscript by adding the peaks was also observed in Sub G1 at 100 and 200 µM.  Also make changes in Lines 385-386.

 Wang YY, Kwak JH, Lee KT, Deyou T, Jang YP, Choi JH. Isoflavones isolated from the seeds of Millettia ferruginea induced apoptotic cell death in human ovarian cancer cells. Molecules. Published online 2020. doi:10.3390/molecules25010207

Chatterjee N, Das S, Bose D, Banerjee S, Jha T, Das Saha K. Lipid from infective L. Donovani regulates acute myeloid cell growth via mitochondria dependent MAPK pathway. PLoS One. Published online 2015. doi:10.1371/journal.pone.0120509

 Author Response Concern 1

As indicated, the suggested correction is now incorporated in the revised manuscript. Please see lines 250-252 and 371.

 Concern 2

 In Line 308-309, The authors have shown Mitogen-activated protein kinase 14 (MAPK14, -5.26-fold) increase. As per the publications. MAPK14 main role could be in protecting cells from apoptosis.

 Liu J, Yu X, Yu H, Liu B, Zhang Z, Kong C, Li Z. Knockdown of MAPK14 inhibits the proliferation and migration of clear cell renal cell carcinoma by downregulating the expression of CDC25B. Cancer Med. Published online 2020. doi:10.1002/cam4.2795

Fang JY, Richardson BC. The MAPK signalling pathways and colorectal cancer. Lancet Oncol. Published online 2005. doi:10.1016/S1470-2045(05)70168-6

 Author Response Concern 2

As indicated, suggested clarification on the downregulation of MAPK14 is now incorporated in the revised manuscript. Please see lines 421-426.